# Modeling the Development of Local Health-Enhancing Physical Activity Policies from Empirical Data and Policy Science Theories

**DOI:** 10.3390/ijerph19031213

**Published:** 2022-01-22

**Authors:** Antoine Noël Racine, Jean-Marie Garbarino, Bernard Massiera, Anne Vuillemin

**Affiliations:** 1French Ministry of Sport, Pôle Ressources National Sport Santé Bien-Etre, 2 Route de Charmeil, 03700 Bellerive-sur-Allier, France; antoine.noel-racine@creps-vichy.sports.gouv.fr; 2Laboratoire Motricité Humaine Expertise Sport Santé (LAMHESS), Université Côte d’Azur, 261 Boulevard du Mercantour, B.P. 3259, CEDEX 03, 06205 Nice, France; jean-marie.garbarino@univ-cotedazur.fr (J.-M.G.); bernard.massiera@univ-cotedazur.fr (B.M.)

**Keywords:** health-enhancing physical activity, municipality, policy science, multiple streams, policy process, policymaking

## Abstract

Physical inactivity is considered a pandemic, requiring strong policy responses to address this major health issue. However, research on the development of Health-Enhancing Physical Activity policies (HEPA) remains scarce, particularly at the local level. There is a need to produce evidence to better understand the process to develop local HEPA policies. This study aims to model the development of HEPA policy promotion in French municipalities from empirical data and policy science theories. This research was undertaken in three steps: (1) assess the level of development of HEPA policies from 10 French municipalities using a local HEPA analysis tool, (2) provide a brief overview of core political science theories applied in health promotion, and (3) from these empirical and theoretical perspectives, model a conceptual framework to better understand the development of HEPA policy promotion in French municipalities. Based on empirical data and the Multiple Streams, policy cycles and Institutional Rational Choice theories, a conceptual framework of the development of municipal HEPA policy promotion was modeled. This conceptual framework is comprised of five stages describing the development of municipal HEPA policies. This paper contributes to a better understanding of the development of municipal HEPA policies.

## 1. Introduction

Insufficient physical activity is associated with major consequences on health, such as an increased risk of developing a non-communicable disease, or a lower quality of life and mental health [1,2,3]. Its prevalence was estimated at 27.5% of the worldwide population [4,5]. In France, 29.3% of the adult population did not meet the World Health Organization [4,5]. Consequently, physical inactivity is considered a pandemic problem [6], requiring strong policy responses to address this major health issue [7,8]. According to the literature, ‘upstream’ approaches such as policies are crucial to increase the physical activity level of the population as part of long-term change [9,10]. To this end, a combination of policy solutions is required, involving many stakeholders from a wide range of sectors at multiple levels to develop a complex ecosystem supporting an active lifestyle [7,8]. Although policy research on health-enhancing physical activity (HEPA) has been expanded over the last two decades [11], more research is needed to produce evidence in order to support policy-makers in their decision to implement HEPA policies [10,12]. Local HEPA policies have a key role to address physical activity issue [13]. In France, the national government transferred some of the legal and ‘facultative’ competencies to municipalities [14]. Thus, municipalities can use their competencies to act on many health determinants such as environmental, social, sport and health factors [15]. Research on HEPA policies at the local level remains scarce [16], unlike the national level [17,18]. A recent review exploring published research on local government policies promoting HEPA showed that few studies analyzed HEPA policies based on theoretical or conceptual frameworks [16]. To fill this gap, the ‘CAPLA-Santé’ framework was recently developed to give an overview and to better understand local HEPA policies in the French context [19]. It provides an overview of these policies and contributes to identifying barriers, levers and remaining challenges for their development. However, using ‘CAPLA-Santé’ [19] does not allow an understanding of how and why local HEPA policies are developed and implemented by policy-makers, as well as why some local governments are more or less involved in HEPA promotion. To translate research into efficient policy and practice, studies should move from the ‘what’ to the ‘how’ [20]. Most of the policy research on physical activity is focused on descriptive studies [21]. For that, theories from political science could enhance the understanding of ‘how’ and ‘why’ health policy are structured and implemented [22,23]. Therefore, a political science approach could help to analyze the process of policy change over time and not just the current content of the policy [22,23]. A political science approach could help to analyze the development process of local HEPA policies in order to provide evidence on policy change to improve their support. As a result, the objective of this study was to model the development of HEPA policy promotion in French municipalities.

## 2. Materials and Methods

This research was undertaken in 3 steps: (1) assess the level of development of HEPA policies from 10 French municipalities using CAPLA-Santé, a local HEPA analysis tool [19], (2) provide a brief overview of core political science theories applied in health promotion, and (3) from these empirical and theoretical perspectives, model a conceptual framework to better understand the development of HEPA policy promotion in French municipalities. 

### 2.1. Assessment of the Development Level of HEPA Policies

To assess different levels of development of HEPA policies, data published in a previous study from the same research project were used [15]. The study analyzed the local HEPA policies in the French Riviera [15]. All mid-size municipalities (between 20,000 and 100,000 residents according INSEE) from the Alpes-Maritimes and Var counties in France (*n* = 10) were targeted. These counties were selected to facilitate data collection being in close proximity to the research team due to the qualitative approach (interviews). Table 1 presents the main characteristics of these municipalities. 

The assessment of the development level of HEPA policies of these municipalities was achieved using CAPLA-Santé [19,20,21,22,23,24], a local HEPA policy analysis tool based on the national HEPA Policy Audit Tool version 2 of the World Health Organization [25,26]. CAPLA-Santé contains 21 items divided into six major sections: overview of HEPA stakeholders in the local government area, policy documents, policy contents, funding and political engagement, studies and measures relating to physical activity in the local government area, and progress achieved and future challenges. The use of CAPLA-Santé is in two steps: a collecting phase and an analyzing phase. Data were collected from each municipality between September 2018 and March 2019 through structured interviews of key informants and the collection of written HEPA policy documents by authors from websites. All responsible or head departments from the sports, health services, and social services sectors of the municipalities were asked to participate to structured interviews. All items of the CAPLA-Santé were addressed to these key informants [24]. Each interview was digitally recorded and transcribed verbatim; The content of HEPA policy documents collected was analyzed using the items and major sections of CAPLA-Santé [24]. Data from each municipality were added to a global matrix including all items of CAPLA-Santé to provide an overview of the progress of HEPA policy development. From this, the research team formulated categories to classify levels of municipality development as following: no policy and no willingness, no policy but willingness, no policy developed but in progress, minor policy or major policy. This categorization process was made from items of CAPLA-Santé. The importance of the policy (minor versus major) was defined on the basis of data collected with the CAPLA-Santé. From items included on the local HEPA analysis tool, eight criteria were selected to establish the importance of the policy: the municipality has written HEPA documents describing the policy, these documents are considered complementary or connected with each other, the policy is intersectoral (involving at least three sectors), the municipality is considered the leader in the promotion of HEPA in the territory, the municipality has a department which ensures cross-sectoral collaboration or coordination in the territory, the policy is translated in concrete actions, the policy has different policy settings and target audiences, the funding and the political commitment of the policy are maintained over several years. If the municipal HEPA policy meets all criteria, it was considered as a major policy. 

### 2.2. Political Science Theories in Health Promotion

A scoping review was led by the research team to provide an overview of core political science theories applied in health promotion. PubMed and Google Scholar databases were used to identify articles published in English and French between 2000 and 2019. The search strategy employed four main keywords: ‘political science’, ‘policy process’, ‘health policy’, and ‘health promotion’. Titles, keywords, and abstracts were screened to select potentially relevant studies on the topic. Then, articles selected from the last step were assessed in full for eligibility. Articles were included when theoretical or conceptual approaches from political science were applied in health promotion or physical activity promotion. The reference list of articles included was also screened in case some research was relevant for this study. Then, main political science theories and approaches were extracted. 

### 2.3. Modeling of a Conceptual Framework

Based on the results of the assessment of the development level of HEPA policies in 10 French mid-size municipalities, the research team selected political science theories which may better explain the policymaking process of these policies, ‘how’ and ‘why’ they were developed. To this end, the authors organized three workshops with the research team to link empirical data and political science theories to model the development of these municipality HEPA policies through a conceptual framework. 

## 3. Results

### 3.1. Assessment of the Development Level of HEPA Policies

A total of 14 written HEPA policies were collected (Figure 1) addressing all items of the CAPLA-Santé in each municipality. 

The analysis of policies content showed that policy settings, target audiences, communication strategies and concrete actions varied among municipalities (Table 2). The development level of HEPA policies was based on this analysis. The municipality ‘F’ had no policy and no willingness (*n* = 1). Municipalities ‘D and J’ had no policy but have willingness (*n* = 2). Municipality ‘E’ had no policy developed but was in progress (*n* = 1). Municipalities ‘B, C, and I’ had a minor policy (*n* = 3). Municipalities ‘A, G, and H’ had a major policy (*n* = 3). 

A total of 21 key informants were interviewed: 10 from the sports, 5 from health services, and 6 from social services departments were interviewed. Key informants reported key moments, strengths, weaknesses, progress, and next challenges of their policies (Table 3). These data reported that the support from national policy, the commitment of elected officials, and large local stakeholder networks facilitated the development of HEPA policies, whereas the lack of intersectoral collaboration and limited resources limited this development. 

### 3.2. Political Science Theories in Health Promotion

The analyzed databases first recorded 4751 articles. A total of 53 articles were screened for full text assessment. Following this last step, 26 articles were included in our review, including three articles relying on political science and physical activity promotion [27,28,29]. From this step, five political science theories were extracted from articles: Multiple Streams [30], the policy cycles [31], Advocacy Coalition Framework [32], Punctuated Equilibrium Theory [33], and the Institutional Rational of Choice [34]. A summary of each theory is presented in Table 4. 

### 3.3. Modeling of a Conceptual Framework

Based on the empirical data and the policy cycles theory [31], five stages of the development of municipal HEPA policies were identified (Figure 2): (0) contextual dynamic flow, (1) agenda setting, (2) policy formulation, (3) policy implementation, (4) policy change.

The Multiple Streams [30] theory was used to explain ‘why’ and ‘how’ HEPA promotion was stated during agenda setting (stages 0 and 1). The Institutional Rational Choice theory [34] was used to explain other stages of the policy development process (stages 2, 3, and 4). The Advocacy Coalition Framework [32] and the Punctuated Equilibrium Theory [33] were not used to model the conceptual framework of the development of municipal HEPA policies. These theories need empirical data over time to be used. 

The conceptual framework of the development of municipal HEPA policies is described in Figure 2. Authors assume that this conceptual framework represents a simplification of a complex process embedded in specific ecosystems. In this framework, each stage of municipal HEPA policy development could be influenced by contextual variables that could be facilitators or barriers, such as the local stakeholders network, the geographic situation, willingness of the mayor etc. [15,35]. Authors considered that contextual variables are not necessarily stable but dynamic. Moreover, authors assumed that the development process of municipal HEPA policies may not always be linear. Thus, the level of HEPA policy development could evolve. 

Stage 0, called ‘contextual dynamic flow’, relied on the Multiple Streams theory [30]. According to Kingdon [30], there are three stream flows: the ‘problem stream’, the ‘policy stream’, and the ‘political stream’. The problem stream is the perception of a specific issue by policymakers such as the level of physical inactivity in the community that requires solutions from the municipality. The ‘political stream’ comprises factors that could influence elected officials, such as beliefs, political sensitivity, electoral intent, legislative constraints, etc. The ‘policy stream’ refers to potential solutions to address the issue, such as physical inactivity. These potential solutions are generally promoted by a ‘policy entrepreneur’ that may be, in this case, local stakeholders, experts, or department heads from municipalities. When two, or all three streams are coupling together, a policy window is opening, giving the opportunity to introduce HEPA promotion in agenda setting (stage 1). However, when none of these streams connect to each other at the same moment, a HEPA policy cannot be developed. In this study, this was the case for municipality ‘F’, which had no policy and no willingness to develop a policy (stage 0). Whereas municipalities ‘D’ and ‘J’ had willingness to develop a HEPA policy and set this topic in their policy agenda (stage 1). In these municipalities, the problem of physical inactivity at the community level was perceived as important by elected officials. Moreover, HEPA promotion was considered as an opportunity to improve the quality of life of their citizens. Elected officials of these municipalities reported that the French national law fostering the prescription of physical activity for people with long-term conditions [36], supported their involvement in HEPA promotion. No policy was reported for municipality ‘E’ but it was in progress, corresponding to the policy formulation (stage 2). In this stage, the municipality explored potential policy solutions and their feasibilities. To this end, the policy solution will be adapted according to rational choices [34], such as the physical and material conditions, the characteristics of the community, possibilities offered by the policy system (legal competencies of the municipality as an authority). However, according to Kingdon’s theory [30], policy entrepreneurs such as experts, local stakeholder networks, or department heads from the municipality could also have important influence on the policy decision and on the choice of the policy solution. Municipalities ‘B, C and I’ had implemented a minor policy (stage 3). In this stage, policy implementation could be achieved with limited resources involving a few departments from the municipality and the policy either has no funding and/or no political commitment over several years. Two situations were distinguished in this stage: the municipality wanted to test and evaluate the efficiency of the policy before further developing it for the long-term; or HEPA promotion was not a high priority topic for elected officials, limiting resources for this policy. For municipalities ‘A, G and H’, HEPA promotion was considered a major policy and important policy change for several years (stage 4).

## 4. Discussion

Based on empirical data and the Multiple Streams [30], policy cycles [31], and the Institutional Rational Choice [34] theories, a conceptual framework of the development of municipal HEPA policy promotion was modeled. This conceptual framework is comprised of five stages describing the development of a municipal HEPA policy from the beginning of the policymaking process to the policy change in the French context. Although this conceptual framework represents a simplification of the whole process of developing municipal HEPA policies, it highlights some mechanisms that could influence policymaking and policy decisions. One of these mechanisms is the perception of the problem such as physical inactivity and the importance for the municipality to lead HEPA policy development. This mechanism described at stages 1 and 2 was also highlighted in previous research [35]. At these stages, strong advocacy could potentially facilitate scaling up the development of HEPA policies. Thus, various stakeholders could play a role in raising awareness of physical inactivity issues to policymakers and to advocate for HEPA solutions [8]. Media, local stakeholder networks, and researchers seemed to play an important role in accomplishing this undertaking [37]. Advocacy seemed particularly important because local policymakers have to deal with other important issues on a daily basis such as economic development, livability, climate change, air quality, natural resource conservation, traffic congestion, traffic safety, etc. [38]. Therefore, advocating for policy solutions promoting HEPA (policy stream) that could be embedded with other issues perceived as important for elected officials (political stream) may foster the chance to open a ‘policy window’ to introduce HEPA promotion in the agenda. At the local level, due to the lack of surveillance data [15], the perceptions of policymakers on physical inactivity as an issue (problem stream) is probably more complicated than at the national level. Pratt et al. [29] showed that physical inactivity is usually perceived as a problem by national governments through a surveillance system from national institutions or research studies. However, having a surveillance system for a municipality is probably not a priority due to budget constraints and limited human resources to operationalize it. Thus, other sub-national governments or local research units could play a role in the surveillance of physical activity levels in order to alert policymakers about this issue (problem stream). 

To model a conceptual framework on the development of municipal HEPA policies, it was necessary not to be limited by the application of the Multiple Stream theory [30]. This policy process theory is mainly focused on agenda setting and disregards subsequent stages of the policy process [39], such as policy formulation and policy implementation. Therefore, the policy cycles [31], and the Institutional Rational Choice [34] were used to explain the following stages of the HEPA policy development. The stage of the policy formulation, when policymakers study possible policy solutions and their feasibility, could be an opportune moment to bridge the gap between research, policy, and practice. If policymakers have access to evidence-based and good practice-based experiences, it will impact the efficacy of policy solutions and interventions [10]. At the moment, research on HEPA policies at the local level remains scarce [16]. In the future, the diffusion of the use of the local HEPA policy analysis tool, CAPLA-Santé [19], in a very large sample of municipalities could provide evidence to policymakers. However, evidence should be easily accessible for them. To this end, establishing a powerful network between researchers and policymakers seems to be promising [37,40]. Once a policy solution is formulated and implementation is decided upon (stage 2 to 3), policymakers will probably face some barriers during implementation [35,41]. A gap could occur between what was planned and how it was implemented. Therefore, it is essential to evaluate the impact of the HEPA policy implemented in the community but also the internal impact within the municipality involved. Thus, the RE-AIM framework [42] assessing five dimensions: reach, efficacy, adoption, implementation, and maintenance at the community and organizational level, may be particularly relevant to evaluate policy implementation and its context. As suggested by Noël Racine et al. [15,35], local contextual factors can affect the development of HEPA policies in mid-size French municipalities. Consequently, contextual variables were taken into account in the conceptual framework presented in this study. However, future research needs to better identify these variables at each stage in the development of HEPA policies. 

This study has some limitations. The empirical data were provided from only mid-size municipalities of two French counties from a short period of time. More analysis of empirical data from a long period of time are missing. The influences of some mechanisms of the local HEPA policies development were not identified. Thus, the generalization of the conceptual framework is limited. Moreover, what is proposed here is a simplification of a complex and dynamic process. Furthermore, three different political science theories were used together in a conceptual way, but it could me more complex to use it to analyse HEPA policies in the real-world. 

## 5. Conclusions

This paper contributes to a better understanding of the development of municipal HEPA policies. The conceptual framework presented in this article allowed for categorization of the level of development of municipal HEPA policies in the French context. Nevertheless, by using empirical data from other countries, this conceptual framework could be adapted and potentially generalizable to other communities across Europe and globally. Moreover, it provides theoretical aspects explaining some mechanisms of the HEPA policymaking process. Applying this conceptual framework could provide evidence on the development of a HEPA policy to facilitate scaling up and support. However, more research is needed to test it in different contextual settings to upgrade this conceptual framework. In particular, each stage should be thoroughly investigated over time in future research. 

## Figures and Tables

**Figure 1 ijerph-19-01213-f001:**
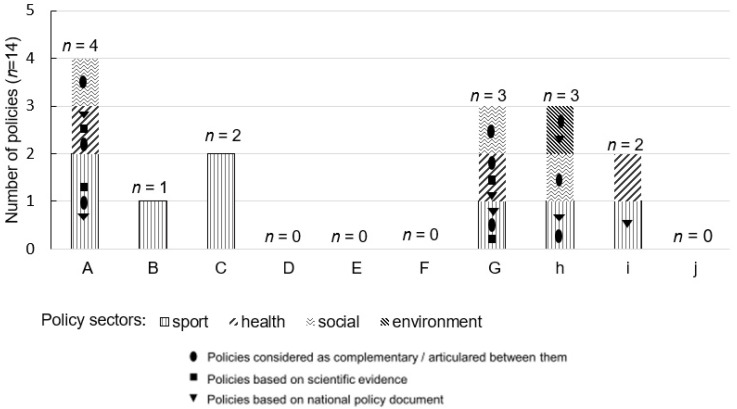
Characteristics of the collected written HEPA policies.

**Figure 2 ijerph-19-01213-f002:**
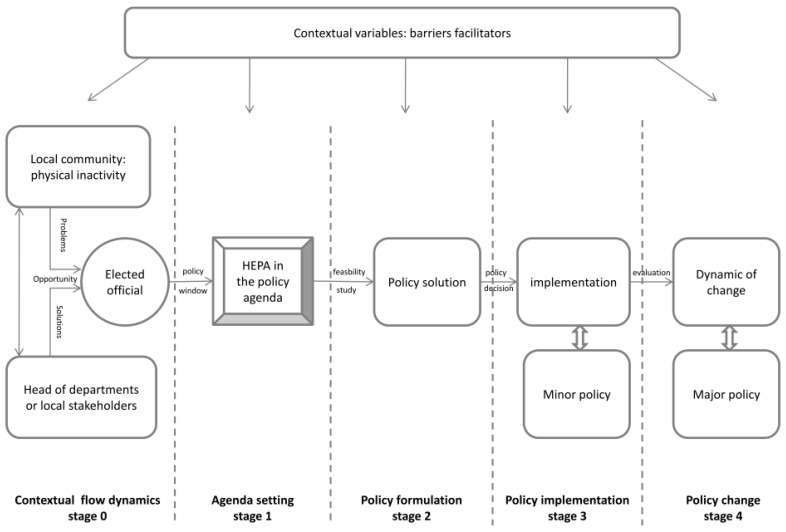
Conceptual framework of the development of municipal HEPA policies.

**Table 1 ijerph-19-01213-t001:** Main characteristics of mid-size municipalities included in the analysis.

Municipality	Inhabitants (*n*) ^1^	Median Income (€) ^2^	People Affected by a Chronic Illness (*n*) ^3^	People Affected by a Chronic Illness (%) ^4^
A	74,875	22,392	12,441	16.6
B	49,322	22,046	8012	16.2
C	28,919	22,858	4592	15.9
D	50 937	20,704	7607	14.9
E	41,571	20,010	7250	17.4
F	35,296	23,152	6913	19.58
G	64,903	18,656	11,305	17.4
H	74,285	18,962	14,369	19.37
I	25,047	20,940	4656	18.5
J	23,347	21,778	3342	14.3

^1^ Data from the National Institute of Statistics and Economic Studies—INSEE (2018). ^2^ Data from INSEE (2018); at national level the annual median income is 22,077 €. ^3^ Number of people affected by a chronic illness covered by governmental insurance for their healthcare expenditure. Data from the Regional Observatory of Provence-Alpes-Côte d’Azur (2018). ^4^ According to INSEE (2018), almost 16% of people are affected by a chronic illness covered by governmental insurance for their healthcare expenditure at national level.

**Table 2 ijerph-19-01213-t002:** Synthesis of the policies content.

Municipality	Policy Settings	Target Audiences	Communication Strategies or Actions	Concrete Actions
A	sports and leisure urban environment health and social care centers	general population seniors inactive and people suffering from chronic diseases	website social networks local newspapers events awareness of healthcare professionals	PA program fitness trails
B	sports and leisure	sedentary people	website events	PA program
C	sports and leisure	seniors	website social networks local newspapers	PA program
D	no data	no data	no data	no data
E	no data	no data	no data	no data
F	no data	no data	website social networks	events promoting PA for heath
H	sports and leisure tourism urban environment	general population inactive people pre-school and children	website social networks events	outdoor fitness and trail network HEPA events
I	sports and leisure primary school priority neighborhoods for urban policy	seniors children	website social networks	PA programs
J	no data	no data	no data	no data

* PA: physical activity.

**Table 3 ijerph-19-01213-t003:** Synthesis of key informants’ feedbacks of their HEPA policies.

Municipality	Keys Moments	Strengths	Weaknesses	Progress	Challenges
A	national legislation of physical activity prescription; local conference on the topic	local stakeholder network; geographic situation; sports facilities; presence of PA and health professionals	limited resources; lack of cycle path network	implementation of HEPA actions	formalize global HEPA action; develop HEPA events; identify recurring funding to sustain HEPA policies
B	pilot implementation of PA program	local stakeholder network; quality and number of PA facilities	lack of intersectoral coordination; geographic difficulties in accessing PA facilities	sustainment of a PA pilot program for seniors to a regular program	improve stakeholder coordination; identify recurring findings to sustain HEPA policies; develop public space to practice PA
C	mayor’s willingness to promote PA	local stakeholder network; geographic situation	few PA programs available for sedentary and inactive people; lack of resources; overuse of PA facilities	implementation of HEPA actions	develop human resources with PA and health training; identify recurring funding to sustain HEPA policies
D	no data	local stakeholder network	unwillingness to develop policy; difficulties moving in the city without a car; lack of resources	no data	build willingness to develop a policy; develop a global intersectoral HEPA policy
E	no data	knowledge of the territory	lack of resources	no data	develop active mobility; develop a HEPA plan
F	no data	local stakeholder network; quality and number of sports facilities; knowledge of the local context	unwillingness to develop policy; lack of knowledge in PA and health, lack of intersectoral collaboration	no data	formalize a HEPA policy; develop a campaign to sensitize residents
G	national policy to promote PA	local stakeholder network; culture of sport; global vision of health	lack of resources and PA facilities	implementation of HEPA actions	develop a global HEPA project from children to older people
H	mandate of the mayor; national HEPA campaign	local stakeholder network, geographic situation; willingness of the mayor; PA facilities	overcrowed PA facilities	implementation of HEPA actions	target more inactive people; develop more cyclable paths
I	national policy to promote PA	geographic situation; PA facilities; good communication	lack of resources	implementation of HEPA actions	target more inactive people; develop a global intersectoral HEPA policy with dedicated human resources
J	no data	local network of stakeholders	lack of PA facilities and lack of public open space	no data	build willingness to develop a policy; develop a global HEPA policy with the metropolis

* PA: Physical activity.

**Table 4 ijerph-19-01213-t004:** Summary of core political science theories used in health promotion.

References	Summary of Theories
Multiple Streams	This theory helps to explain (1) why some issues are stated in the political agenda through policy dynamics and (2) how a policy window could lead to policy change. Kingdon distinguishes 3 stream flows: the problem stream, the policy stream, and the political stream. When these streams come together, a window policy is opened providing a possibility to make changes in the policy.
Policy cycles [31]	This theory allows for a better understanding of the policymaking process using a heuristic method to describe main stages of policy cycles. Howlett et al. identified 5 main stages in the policy cycle: agenda setting, policy formulation, policy decision, policy implementation, and policy evaluation.
Advocacy Coalition Framework [34]	This theory assumes that the policymaking decision leading to policy change is embedded in the policy system by the interaction of advocacy coalitions. A strong alignment of the advocacy coalitions over time is needed to drive policy change.
Punctuated Equilibrium Theory [32]	This theory suggests that policymaking is characterized by long periods of stability (equilibrium) with minor policy change, punctuated by brief periods of major change. According to Baumgartner and Jones, the process of change is embedded in a complex system and is linked to the policy image from public opinion and the involvement of a set of stakeholders in a particular issue.
Institutional Rational Choice [34]	This theory argues that institutions make rational choices to maximize achieving their objectives. There are 3 main categories of factors that influence the institutional choice: the physical and material conditions, the characteristics of the community, as well as the rules and the policy system.

## Data Availability

The data that support the findings of this study are available from the corresponding author, upon reasonable request.

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
