# Peer review of "Modeling the Development of Local Health-Enhancing Physical Activity Policies from Empirical Data and Policy Science Theories"

_ijerph, 2022, doi:10.3390/ijerph19031213_

Round 1
Reviewer 1 Report
Dear Authors,
Thank you for this article, which fills an important gap in the promotion of physical activity and makes an important contribution to the substantially underdeveloped health policy analysis.
Attached are some suggestions for revising the article:
Background:
Line 36: Is the concept of sustainability being addressed here or is this about long-term change? In terms of terminology, this should be sharpened.
Line 38-40: Key research findings should be briefly cited before referring to the gap.
Line 45: A brief description of the "CAPLA-Santé" framework should be given. A detailed description is given later in the methods section, as is correct.
Line 50-51: The conclusion should be supported by arguments.
A short classification to the legislation (national, municipal) level in the context of health promotion should be added.
Methods:
A short classification of the selected regions should be made, e.g. are they representative for France. Also, it should be explained why these regions were selected.
It is worth considering entering the national comparative values for France in Table 1.
Line 82-84: How was the "CAPLA-Santé" framework developed and for which areas of application is it valid? What empirical evidence is available regarding its methodological quality?
Lines 87-89: How were the key informants selected? What was the sampling strategy? How were the data collected? How were the data analyzed? Important details are missing here and should be included.
Line 91-93: Were the categories developed for this analysis or are these the categories of the "CAPLA-Santé" framework? How was the classification into these categories made?
Line 96: Was it only recorded whether documents were available or were they also analyzed? This should be presented in a structured way in the method description: 1) structured interviews and 2) document analysis (if it is so correct). Then, for each phase, it should be described in detail how they were conducted: from data collection to data analysis.
Line 103: What other classifications are there? (see results section) These should be explained here.
Line 106 and following: How were the items selected? Was this done by one person or by several people?
Line 120: How were the political science theories selected?
Results:
Line 161 and following: The use of the different theoretical approaches should be justified. The model seems very general - only in one place physical inactivity is referred to. It is unclear how and in what way the results of the first empirical step were used. This should be highlighted.
Line 170-171: Which facilitators and which barriers are meant? These should be specified.
Section 3.1: The results obtained from the interviews are missing in the results section. These are essential, however, for the conclusion drawn in the discussion.
Discussion
In reference to the question posed in the results section, is it a specific or general model, should be addressed in a differentiated manner in the discussion section. I.e. where are specifics and where are generalizations possible. Furthermore, it would be interesting to discuss the differences in influence depending on the role of the stakeholder, e.g., for agenda setting. In addition, the influence of contextual factors should be taken up.
The role of the media is only addressed in the discussion. Where does this play a role in the results?
Line 220: Highlighting relevant mechanisms is very interesting. The potential of the results should be presented more broadly here. E.g., What other mechanisms are effective; how do the mechanisms influence each other (reinforce, reduce)?
Line 261 and following: Why is RE-AIM referenced here without considering it for the study as a whole.
Line 269: The limitations are very brief and do not cover all critical points; in particular, it should be reflected that very different theories are used and a theoretical assessment of this compilation and its limitations should be made.
Reviewer 2 Report
This paper did an excellent job of integrating five theories from political science to explain the development of HEPA policies. My main concern is to link the qualitative studies from 10 municipalities with the theoretical framework (Figure 2).
In the discussion section, the authors discussed the 5 stages, but didn’t show much evidence from the 10 municipalities. Authors need to show the information from policy files and interviews to support the arguments. For example, what kinds of policies are used to improve physical activity. What were the interview questions the authors ask? What can we learn from the interviews? As the authors stated that municipality F had no policy and no willingness to develop a policy, how did the interviewees from the departments of sports sector and health sector respond to that? The authors have very rich information on the issue, and it should be shown in the paper.
In the discussion section, the authors stated the importance of civic engagement in the development of HEPA policy. For example, F, D, and J are developing the HEPA policy, and it could be related to the awareness of elected officials and civic engagement. If the professionals and civic engagement matter, could those factors help municipality F to move forward? Municipalities A, G, H have policies on multiple sectors (Figure 1). Authors could use the framework (Figure 2) to explain how different stages push the HEPA policies in those communities. Will multi-sector collaboration matter and how does the collaboration work among different sectors?
In the discussion section, as authors come up with the framework from political science, what will be the policy implications from the framework? I was wondering if the interviewees talked about the barriers of policy design and policy implementation, and how they overcame the barriers.
Other comments:
Section 2.1. can authors include more detailed information on the selection of 10 municipalities? Although authors cited [30], it will be useful to include more information here.
In Table 1, it will be helpful to show the % of people affected by chronic illnesses and additional illnesses. Do authors have the data on physical inactivity? Also, how does the % of chronic illnesses compare to the national level? If the authors have the information, it will be helpful to show the changes of chronic illness over time to see if the policy makes a difference.
Figure 1 is very blurry
The CAPLA-Sante framework should be briefly described.
Table 2 could be summarized into sentences.
Reviewer 3 Report
Very interesting work. However, some important elements are missing:
Description of the state policy in the field of physical activity and promotion programs
Failure to specify what direction of the municipal policy is assessed, whether they are programs for seniors, school, general ...
No reference to other countries or regions
Speaking of the political determinants of programs aimed at physical activity at work, there is no reference to EU WHO guidelines or legal conditions applied in a given region
Reviewer 4 Report
The article provides a good abstract helping the reader to have a proper idea of the research question and the aim of the study but I would rephrase the title to make it clearer.
I think that this paper is very interesting and useful to understand the mechanisms underlying the processes of development of a HEPA at local level. However, due to some simplification of the analysis in the paper (mentioned also by the authors), it would be difficult to apply the model in other contexts without additional information.
Here you can find more specific comments:
BACKGROUND:
In my opinion the background section could be further improved with more recent references and a better reasoning of the importance and originality of this research.
Moreover, information about PA data in France could be also useful to better understand the context.
References 1-2 are a little old
Reference 3: add also ref. with update data from WHO
Line 42 : add some recent reference on HEPA assessment at National level (for example Gelius, P., Messing, S., Forberger, S., Lakerveld, J., Mansergh, F., Wendel-Vos, W., ... & Woods, C. (2021). The added value of using the HEPA PAT for physical activity policy monitoring: a four-country comparison. Health Research Policy and Systems, 19(1), 1-12).
METHODS:
Line 63: Even if you give the reference of Capla Sante tool, I would write more details here.
Section 2.1 lines 82-105: maybe would be useful for the reader to see a graphical representation of the sections and categories.
Table 1: could you add data on PA levels of these specific populations? Do you have this data?
Section 2.2: is the review Systematic or not?
Section 2.3 line 122: explain what you mean for workshop and who participated.
RESULTS
Figure 1: I would avoid colours for data visualization (just in case people decide to print the article).
Table 2: In my opinion these information can be additional material but the table doesn’t need to be part of the main paper. You could add some details in the text.
DISCUSSION
The discussion is very well written and the findings are properly described in the context of the cited literature
CONCLUSION
In my opinion it may be worthwhile to discuss more detailed ideas for further investigation and practical implication of the application of the model to other contexts.
Round 2
Reviewer 2 Report
The paper has been improved. The new table 2 and table 3 are very informative. I suggested authors summarize some key points and describe both tables in the text. For example, table 3 shows that the local stakeholder network is the main strength. The weakness includes the lack of resources, motivation, and intersectoral collaboration.
There are two Table 3 in the text.
Author Response
The authors would like to thank the reviewers for the time they spent reviewing this paper and their comments. We sincerely appreciated your input that helped to improve the manuscript. The authors have followed reviewer’s suggestions.
Best regards,
The authors
The paper has been improved. The new table 2 and table 3 are very informative. I suggested authors summarize some key points and describe both tables in the text. For example, table 3 shows that the local stakeholder network is the main strength. The weakness includes the lack of resources, motivation, and intersectoral collaboration.
Some key points have been added from table 2 and 3 in the text:
“The analysis of policies content showed that policy settings, target audiences, communication strategies and concrete actions varied among municipalities (Table 2). The development level of HEPA policies was based on this analysis” (lines 150-153).
“These data reported that the support from national policy, the commitment of elected officials, and large local stakeholder networks facilitated the development of HEPA policies, whereas the lack of intersectoral collaboration and limited resources limited this development » (lines 166-168).
There are two Table 3 in the text.
A modification has been made (lines 182-1). Table 3 has been replaced by Table 4.